# Antimicrobial Activity Developed by Scorpion Venoms and Its Peptide Component

**DOI:** 10.3390/toxins14110740

**Published:** 2022-10-28

**Authors:** Clara Andrea Rincón-Cortés, Martín Alonso Bayona-Rojas, Edgar Antonio Reyes-Montaño, Nohora Angélica Vega-Castro

**Affiliations:** 1Grupo de Investigación en Productor Naturales, Facultad de Ciencias, Universidad De Ciencias Aplicadas y Ambientales U.D.C.A, Calle 222 No 55-37, Bogotá 111166, Colombia; 2Grupo de Investigación GIBGA, Facultad de Ciencias de la Salud, Universidad De Ciencias Aplicadas y Ambientales U.D.C.A, Calle 222 No 55-37, Bogotá 111166, Colombia; 3Grupo de Investigación en Proteínas, Facultad de Ciencias, Universidad Nacional de Colombia, Sede Bogotá, Av Cr 30 No 45-30, Bogotá 111321, Colombia

**Keywords:** venom, scorpion toxins, peptides, antimicrobial, structural properties, Buthidae family

## Abstract

Microbial infections represent a problem of great importance at the public health level, with a high rate of morbidity-mortality worldwide. However, treating the different diseases generated by microorganisms requires a gradual increase in acquired resistance when applying or using them against various antibiotic therapies. Resistance is caused by various molecular mechanisms of microorganisms, thus reducing their effectiveness. Consequently, there is a need to search for new opportunities through natural sources with antimicrobial activity. One alternative is using peptides present in different scorpion venoms, specifically from the Buthidae family. Different peptides with biological activity in microorganisms have been characterized as preventing their growth or inhibiting their replication. Therefore, they represent an alternative to be used in the design and development of new-generation antimicrobial drugs in different types of microorganisms, such as bacteria, fungi, viruses, and parasites. Essential aspects for its disclosure, as shown in this review, are the studies carried out on different types of peptides in scorpion venoms with activity against pathogenic microorganisms, highlighting their high therapeutic potential.

## 1. Introduction

Many microorganisms, such as bacteria, fungi, viruses, and parasites, have been reported as resistant to multiple antibiotics, annual deaths worldwide are expected to increase from 700,000 in 2014 to 10 million in 2050 [1,2,3]. For example, *P. aeruginosa* resistance has developed intrinsically, being responsible for 10% of all hospital-acquired infections worldwide. Concerning *Staphylococcus*, a common colonizer of the skin and mucous membranes, many species are opportunistic pathogens, representing a threat to human health due to their broad resistance. Methicillin-resistant *S. aureus* (MRSA) has been widely implicated in various diseases such as pneumonia and skin and bloodstream infections. It is associated with multiple healthcare-acquired diseases worldwide. Considering this rise of infection mentioned above, it is vital to investigate complementary treatment options to combat these pathogens, using antimicrobial peptides (AMPs). Which are important peptides in the immune system of different organisms and have growth inhibitory activity in bacteria, fungi, parasites, and viruses. Some of the AMPs have been identified in scorpion venom [4,5,6,7].

Scorpions are invertebrate animals, recognized by their antiquity, belonging to the Arthropoda type, Chelicerata subtype, Arachnida class, and Scorpionida order. Twenty families are currently recognized [8]; eight families stick out due to their wide worldwide distribution, known as Bothriuridae, Buthidae, Chactidae, Chaerilidae, Diplocentfidae, Iuridae, Scorpionidae and Vaejovidae [9,10]. Of these families, the Buthidae stands out since some of the genera and species belonging to this family manage to develop different toxicity levels for mammals and insects. In addition, their activity in growth inhibition of cancer-derived cells and antimicrobial activity has been studied; characteristics mainly attributed to the protein components found in the venom of scorpions belonging to Buthidae family and the other families. This paper reviews the peptides in scorpion venoms belonging to the Buthidae family mainly, which possess or have been reported to have antimicrobial activity, highlighting their activity on different types of microorganisms.

## 2. Antimicrobial Peptides (AMPs) in Scorpion Venoms

Scorpionism represents a public health problem, one of the most common causes of toxicological consultation in some regions of the world [11]. More than 1,200,000 accidents are reported annually, which result in 3250 deaths [11,12,13,14]. The scorpion employs venom for defense or attack purposes to seek food. This venom contains many components, such as mucoproteins, lipids, enzymes, inorganic salts, nucleotides, amines, and peptides. About 80 toxins modulate ion channels and receptors’ function in excitable cell membranes, responsible for the multiple known symptoms of envenomation and isolated bioactive peptides with antimicrobial effects. Less than 1% of the estimated 100,000 peptides produced by scorpions are known. With natural selection, different types and subtypes of receptors (ion channels) co-evolved in various animal groups; coincidently, specific ligands related to peptides with toxin activity evolved in scorpions [15,16,17]. Toxins produced by scorpions have attracted the scientific community’s attention for their therapeutic properties; an example is the Cuban blue scorpion *Rhopalurus junceus*, which is used in oncologic therapy and other species for having antimicrobial activity against bacteria and fungi [18,19]. AMPs are amphipathic with a positive charge. Structurally categorized into peptides with disulfide bridges, an amphipathic α-helix without cysteine residues, and rich amino acids such as proline and glycine [20]. Additional molecular weights from 1.5 to 8.3 kDa, basic nature, among which the pandinins of *Pandinus imperator* [21], the hadrurin of *Hadrurus aztecus* [22] and the parabutoporin of *Parabuthus schleteri* [23] stand out.

The AMPs are considered natural antibiotics in unicellular organisms and mammals [24], with a broad spectrum of activity against fungi, bacteria, viruses, and protozoa [25]. Further, these peptides have a critical protective action in organisms lacking an immune system. In vertebrates, they are part of innate immunity; likewise, they are not associated with resistance to classical antibiotics [26]. Therefore, AMPs’ diverse structures are positioned as potential therapeutic elements to be characterized and employed in developing new-generation antimicrobial drugs, such as IsCT from the scorpion *Opisthacanthus madagascariensis*, BmKn2 from *Buthus martensii* Karsch, Pandinins and Scorpine from *Pandinus imperator* and Vejovine from *Vaejovis mexicanus*, all with a series of diverse structures, positioned as potential therapeutic elements to be characterized and employed in the design and development of new-generation antimicrobial drugs [20,27,28,29,30,31].

Lately, AMPs studies are of great relevance, being recognized as a therapeutic alternative against various microorganisms. Additionally, they have a low association with antimicrobial resistance due to their mode of action, which is associated with damage to cell membranes and at the level of intracellular targets capable of blocking metabolic pathways and important biological functions for the development of microorganisms. Furthermore, antimicrobial activity is fast (within minutes) and highly effective (99.9% lethality), whereas most antibiotics have slower mechanisms of action.

Mechanisms of AMPs’ resistance are related to changes in the molecular composition of the microbial membranes, moreover, a decrease in the electrostatic activity is necessary for the interaction of the peptide with the plasma membrane; similarly, some efflux pump systems and protease expression have been described as strategies to increase resistance to AMPs [32,33,34]. Among the mechanisms of action of AMPs, some have antimicrobial activity, while others have an indirect effect by activating the innate immune system. In addition, the differences in the cell membrane composition between the pathogens and the host cells have underpinned the targeting specificity of AMPs. Some cell wall constituents are common to Gram-positive and Gram-negative bacteria, such as peptidoglycans. In contrast, other cell wall components are specific to either Gram-negative microbe. For example, lipopolysaccharide (LPS), also called endotoxin, is found exclusively in Gram-negative bacteria. At the same time, lipoteichoic acid (LTA) is specific to Gram-positive bacteria, the latter increasing the membrane’s negative charge. Unlike bacteria on fungal membranes, negative charge mainly results from the phosphomannan and 1,3-β glucan. Other compounds are diphosphatidylglycerol (DPG), cardiolipin in bacteria, and phosphatidylinositol (PI). In addition, some AMPs have intracellular targets and perform antimicrobial activity by inhibiting protein synthesis and DNA replication [35,36,37].

### 2.1. Peptides with Antibacterial Activity (ABPs)

Scorpion antimicrobial peptides have been classified into long and short-chain; its N-terminal is an alpha-helix without disulfide bridges, and the C-terminal domain is shaped by three disulfide bridges, similar to the sequences of potassium channel toxins. The first scorpion toxin was isolated from the venom of *Pandinus imperator*, composed of 75 amino acids. It showed antimicrobial activity against *Bacillus subtilis* with a minimum inhibitory concentration (MIC) of around 1 μM and against *Klebsiella pneumoniae* with a MIC of approximately 10 μM [38].

Antibiotics have become widespread, and the rise of their use has been directly associated with an increase in bacterial strains resistant to them, becoming a global problem. Also, many antibiotics have been used in animal feed, worsening such resistance. Currently, efforts are directed toward researching new generations of antibiotics with new mechanisms of action [39,40]. AMPs show broad-spectrum activity against a diverse group of Gram-positive and Gram-negative bacteria; very few have been characterized to date, with approximately 25 compounds evaluated [20,41,42]. However, it has been established that they are less likely to develop resistance and have an advantageous effect on nutrient sources and gut microbiota in animals [43,44,45].

Guo X et al., 2013 [46] reported two, TsAP-1 and TsAP-2, whose structures were deduced from cDNAs cloned from a venom-derived cDNA library of the Brazilian yellow scorpion, *Tityus serrulatus*, which were structurally characterized. TsAP-2 was highly potent against the Gram-positive bacterium, *Staphylococcus aureus* (MIC 5 μM), while TsAP-1 was low potency (MICs 120 μM). Additionally, substitutions of four neutral amino acids with Lys residues in both peptides increased their activity. Also, it was demonstrated that the venom of *Veajovis punctatus* presented an antimicrobial effect to inhibit *S. aureus* effectively and *Streptococcus agalactiae* at different concentrations. Besides, the antibacterial capacity of venoms of *Scorpions tibetanus* and *T. serrulatus* against Gram-positive bacteria was studied. The peptide Vejovine, isolated from the venom of *Vaejovis mexicanus*, showed potent antimicrobial activity against Gram-negative and multiresistant clinical strains to antibiotics, such as *E. coli*, *P. aeruginosa* and *Acinetobacter baumanii*, with MIC values of around 4.4 μM [28].

Some scorpion species’ venoms possess antimicrobial effects against Gram-positive and Gram-negative bacteria, such as *Enterococcus faecium*, *Streptococcus agalactiae*, *Micrococcus luteus*, *Staphylococcus aureus*, *Escherichia coli*, *Pseudomonas aeruginosa* and *Enterobacter cloacae* [47,48,49]. A new class of antimicrobial peptides was identified in species of families other than the Buthidae such as, the venom gland of *Heterometrus spinifer*, named HsAp, HsAp2, HsAp3, and HsAp4, respectively. Each of the four peptides contained 29 amino acid residues, with characteristics of being cationic and weakly amphipathic. No significant homology with any other peptide was evident; therefore, they are a new peptide family. Antimicrobial analysis showed that HsAp inhibits the growth of Gram-negative and Gram-positive bacteria with MIC values between 11.8–51.2 Μm [47], identified Hp1404, a novel cationic antimicrobial peptide from the scorpion *Heterometrus petersii*, which exhibits specific inhibitory activity against methicillin-resistant *S. aureus*. Hp1404 peptide can penetrate the bacterial membrane at low concentrations and disrupts it at very high concentration. While most isolated and characterized venom components are peptides, two 1,4-benzoquinone compounds from scorpion’s venom *Diplocentrus melici* showed remarkable antimicrobial activities against *Staphylococcus aureus* and *Mycobacterium tuberculosis*, respectively. Further, the blue compound is equally effective against normal and multidrug-resistant tuberculosis and does not affect the epithelium of the lungs, heightening its potential as a new drug candidate [50]. Two cationic peptides were isolated from *Liocheles australasiae* venom (Lausporin-1 and Lausporin-2); which exhibited antibacterial (plasma membrane disrupting) activities against methicillin-resistant *Staphylococcus aureus*, *S. epidermidis* and *S. capitis*, at minimum inhibitory concentrations ranging from 2.5 to 10 μg/mL [51]. In addition, the antibacterial capacity of six peptides obtained from *Mesomexovis variegatus* scorpion venom was evaluated; these peptides, belonging to the NDBP-4 family, were structurally and functionally characterized. All six peptides showed hemolytic activity, inhibiting the growth of *S. aureus*, but were not effective against *Pseudomonas aeruginosa* [52]. There are 67 ABPs from scorpion *Opisthacanthus madagascariensis* venom with antimicrobial and low hemolytic activity [53,54]. The antibacterial activity of three compounds obtained from *Urodacus yaschenkoi* venom was just documented. Peptides with alpha helix (α-helix) structure have bactericidal activity against β-hemolytic *Streptococcus* and low hemolytic activity. Therefore, represent an alternative for developing new antimicrobial drugs [55].

Peptides from *Hadruroides mauryi* [56] and *Centruroides margaritatus* (Buthidae) [57] venoms are similar and isolated using a cation exchanger on CM Sephadex C-25 at pH 7 from which seven fractions were obtained, which inhibited the growth of *E. coli*, *P. aeuginosa*, *B. cereus*, and *S. aureus*. Peptides had a basic character and did not show hemolytic activity [58]. A peptide with antibacterial activity from the venom of *C. margaritatus* [57], represents 3% of the total venom protein. By -SDS-PAGE, it was determined to have a molecular weight of 7.3 kDa and inhibited the growth of *B. cereus*, *S. aureus*, *P. aeruginosa* and *S. marcencens*, in microplates with Davies minimal medium. Nevertheless, the Müller-Hinton agar assay did not show significant inhibition halos; consequently, the peptide has bacteriostatic but not bactericidal activity. Likewise, it did not affect *E. faecalis*, *E. coli*, *S. choleraesuis*, or *K. pneumoniae* [59]. The venom of the scorpion *Androctonus australis* (Fat-tailed scorpion) displays inhibition against different bacteria through membrane-level damage of microorganisms. Proteomic analysis revealed that molecule to be an inhibitor of sodium channels. Among the bacteria inhibited were *Bacillus cereus*, *S. aureus* ATCC 25923, *Micrococcus sp.* and *Escherichia coli* ATCC 25922; this scorpion belongs to the family *Buthidae* and inhabits North Africa and South and West Asia.

Antibacterial activity in vitro of venom obtained from *Hadruroides charcasus* (Karsch, 1879) showed that the *P. aeruginosa* strain was more sensitive (MIC 0.07 mg/mL) than *S. aureus* (MIC 0.035 mg/mL); molecular weights around 7.0, 8.3, 8.3, and 9.1 kDa were determined for some peptides [60].

The peptide, BmKn, from the venom of the Chinese scorpion *Mesobuthus martensii* (Karsch, 1879), has higher microbial activity against Gram-positive bacteria [61]. Also, fractions from *Centruroides tecomanus* (Hoffmann, 1932) scorpion venom possess antibacterial activity [62] on *S. aureus* and *P. aeruginosa*. Moreover, peptides denominated AamAP1, and AamP2 peptides from the North African scorpion *Androctonus amoreuxi* (Audouin, 1826) [63] had higher activity against *S. aureus*. Similarly, the venom of *Vaejovis mexicanus* (Koch, 1836) was more effective against Gram-negative bacteria, such as *P. aeruginosa* including multidrug-resistant strains [28]. Certain species, such as *Heterometrus laocticus*, *Tityus discrepans and Opistophthalmus carinatus*, produce peptides with antimicrobial effects called defensins [64,65,66]; other related peptides, such as Opistoporin-1 (*Opistophthalmus carinatus*) and Pandinin-1 (*Pandinus imperator*), both with 44 amino acids, have antimicrobial activity against Gram-positive and Gram-negative bacteria, respectively [21,67].

Hadrurin peptide from *Hadrurus aztecus* had bactericidal activity against *E. coli*, *Enterococcus cloacae*, *Klebsiella pneumoniae*, *Salmonella typhi*, and *P. aeruginosa* [22]. Similarly, the peptide BmKn2 from the venom of the Chinese scorpion *Mesobuthus martensii* (Karsch, 1879) exerts great activity against Gram-positive bacteria [62]; finally, a peptide obtained from the Asian scorpion *Scorpiops tibetanus*, called StCT2, showed an antibacterial effect against *S. aureus*, including methicillin-resistant strains [27,61]. The minimum inhibitory concentrations (MICs) were 6.25–25 μg/mL; the last findings show the therapeutic potential of StCT2 as a novel amicrobial peptide. Table 1 shows several peptides isolated from scorpion venoms and the activity that has been related to each one.

### 2.2. Peptides with Antifungal Activity (AFPs)

Antifungal AFPs are linear and cyclic molecules, present amphipathic or hydrophobic properties, and exhibit the activity of lysis, the binding, and rupture of a cell’s outer membrane. In addition, some peptides manage to bind with different proteins of the nuclear envelope of certain species of fungi, producing reactive oxygen species, ion outflow, and ATP, thus, causing cell death [71]. Other mechanisms are related to the alteration of surface membrane tension, causing the formation of pores and the release of K^+^ and another series of ions, as well as acting as mitochondrial regulators [72]. The AFPs can develop both fungicidal and fungistatic activity and have simultaneous activity against bacteria. One of the examples of AFPs is Cm-p, described in studies carried out by [73]. It was shown that they have in vitro fungistatic activity against *Candida albicans*, an opportunistic yeast, with a minimum inhibitory concentration of 10 µg/mL. In contrast, other AFPs show great activity against pathogenic fungi such as *Fusarium oxysporum*, *Botrytis cinerea*, *Aspergillus niger*, *Cryptococcus* spp., and *Saccharomyces* spp. [74,75].

Regarding fungi, an increase in resistance and mortality caused by *Candida* and *Aspergillus* has been established [76]. Candidiasis is responsible for 90% of invasive mycoses worldwide, causing different clinical manifestations, generated by different species of Candida as *C. albicans*, *C. auris*, *C. glabrata*, *C. parapsilosis*, *C. tropicalis* and *C. krusei* [77,78]. That can affect immunosuppressed patients, including those undergoing chemotherapy treatment, with hematological neoplasms, or in intensive care [79,80].

Thus, from this problem, some researchers evaluated the in vitro antifungal activity of the diluted crude extract of scorpion venom *Hadruroides charcasus* (Karsch, 1879) against *C. albicans* using 54 experimental units (nine venom concentrations, two strains, and three replicates) and the microdilution method, demonstrated that this venom possesses antifungal activity against *C. albicans* [81]. Several studies have emerged on the antifungal activity of the venom of different species of scorpions or even their peptides, as is the case of toxinological studies of the peptides from venoms for scorpions of the Vaejovidae family, for example, VmCT1 and VmCT2 of *Vaejovis mexicanus smithi* [34,35], VpAmp1.0 y VpAmp2.0 of *Mesomexovis variegatus* [22]. It is possible to observe the great variety of peptides that inhibit the growth of different fungi, in addition to having specific MIC values that could be proposed as pharmacological agents for treating diseases caused by this type of organism, Table 2 shows a summary of peptides with antifungal activity.

### 2.3. Peptides with Antiviral Activity (AVPs)

Antiviral peptides (AVPs) can be an alternative for managing diseases such as influenza, dengue, hepatitis C, herpes simplex, and HIV. For treating some diseases caused by viruses, conventional antiviral therapies are currently used, but these are limited, even in some of them, there is no treatment; this in addition to the development of resistance of different viruses to treatments explicitly developed or generally. AVPs can be classified within antimicrobial peptides and directly identify the virus because they act directly on viral envelopes, glycoproteins, and capsids. At the same time, other compounds bind to cell receptors preventing their interaction with the virus or inhibiting its replication [88]. They can also activate the host immune response and thus inhibit one or more stages of the virus life cycle [89,90]; there is a great variety of AVPs, for example, the α-defensins HNP-1, 2 and -3 and HD-5, with which in vitro studies managed to block the infection caused by the papillomavirus [68]. Some of the AVPs present in the venoms of different species of scorpions are classified as peptides with disulfide bridges (DBPs), which allows them to bind with the glycoprotein gp120 of the human immunodeficiency virus (HIV), thus preventing the interaction between gp120-CD4, accordingly avoiding the virus from entering to the host cell [91,92]. This mechanism is a “camouflage” of CD4 with DBPs since the tertiary structure of the peptide resembles the CD4 CDR2 loop [91]. The toxins charybdotoxin (ChTx) and scyllatocin, purified from *Leiurus quinquestriatus* scorpion venom, have a CS-alpha (α)/beta (β) motif responsible for blocking K^+^ channels [91,93,94,95,96,97], a helpful structure for studying the interaction between gp120-CD4 [91,98,99]. Other peptides related to simian (SIV) and HIV antiviral activity are obtained from cDNA analysis of the venom gland of *Tityus obscurus*, *Opisthacanthus*
*cayaporum*, *Hadrurus gertschi*, and *Lychas mucronatus* [100,101]. One of the peptides of *T. obscurus* is highlighted, with 26 amino acid residues, which developed activity against SIV by activating the immune response of the cell and neutralizing the virus and the peptides mucroporin, mucroporin-M1 and mucroporin -S1 of *L. mucronatus*, with high anti-HIV-1 activity developing an IC50 of 1.65 μM [101,102]. Another of the peptides with activity on this same virus (HIV-1) is the BmKn2 cloned and synthesized from *Mesobuthus martensii* cDNA, with anti-HIV-1 activity, by inhibiting the activities mediated by the chemokine receptors CCR5 and CXCR4 and virus replication [91,102]. An antiviral study using venom from different scorpion species against the HCV virus showed that *Scorpio maurus palmatus* and *Androctonus australis* had an activity with IC50 of 6.3 and 88.3 μg/mL, respectively. Therefore, *S. maurus palmatus* was proposed as a natural product and a great anti-HCV agent [91,103]; also, *Liocheles australasiae* has enzymes such as phospholipases (LaPL_2_-1) with activity in viruses such as hepatitis C and others [101,104]. Other peptides have activity on several viruses such as hepatitis C (HCV), dengue (DENV), measles, influenza (HN), Zika (ZIKV), herpes simplex (HSV), Japanese encephalitis (JEV), and SARS-CoV (Table 3).

### 2.4. Peptides with Antiparasitic Activity (APPs)

Parasitic infections in humans are increasing due to the appearance of strains resistant to antiparasitic compounds, so the search and development of new compounds have been carried out, focusing on the extraction and purification of active principles of natural or synthetic origin from the venom of different species of animals [112,113,114], as is the case of scorpion venom which has deomnstrated the leishmanicidal effect (on *L. mexicana*). Borges, et al., 2013 [114] conducted a test from twelve venoms of ten species of scorpions of the genus *Tityus*, endemic to Venezuela, and one Brazilian species (*T. serrulatus*). The following species are classified depending on their antiparasitic activity; high activity (>80% mortality; *T. discrepans*, *T. gonzalespongai* and *T. perijanensis*), intermediate activity (40–80% mortality; *T. clathratus*, *T. falconensis A*, *T. nororientalis AL*, *T. nororientalis CA*, *T. imei and T. zulianus*), and low activity (<40%; *T. falconensis B*, *T. breweri*, *T. sanarensis*, *and T. serrulatus*). Among the relevant results, it was documented that from the venom of *T. gonzalespongai*, a component of 6880.97 Da named HPLC-13 was isolated, which proved to be a potent anti- leishmanicidal in vitro assay. Parasite inhibition is due to the collapse of the *L. mexicana* morphology, with vacuolization and loss of the flagellum structure [114]; the VmCT1 peptide from venom *V. mexicanus* inhibits *Trypanosoma cruzi* with EC_50_ of 1.37 µmol L^−1^ [115]. Other examples are shown in Table 4.

## 3. Discussion

Antimicrobial peptides widely exist in nature and are components of the innate immunity of almost all living things. They have many antimicrobial activities against bacteria, fungi, viruses, parasites, and other microorganisms and play an essential role in resisting foreign invading microorganisms. As mentioned previously, many of them are an alternative to traditional antibiotic-resistant strains and do not quickly induce the development of drug resistance. Though these peptides present a new opportunity, much remains to be learned until a new drug is developed. The activity of AMPs occurs through complex mechanisms of action, including acting on the cell wall, cell membrane, and different intracellular targets, as well as antibiofilm formation and host immune system modulation activities (Figure 1).

The differences in the cell membrane composition between the pathogens and the host cells strengthen the targeting specificity of AMPs. In turn, the difference in composition of the membranes of diverse pathogens increases its specificity and effect. Mechanisms involved in the antimicrobial activity are related to their hydrophobicity and electrostatic interaction of cationic peptides with anionic surfaces; it makes the insertion of peptides in the cell wall more accessible, the initial step in the antibiotic effect. The action of AMPs and the specific cellular targets can vary among peptides and organisms. Therefore, a well-defined structure is mandatory for antimicrobial activity. For example, it is common to have an α-helix or a β-Sheet and lack of disulphide bonds in AMPs as well as the presence of both hydrophobic and hydrophilic regions.

AMPs studies are ongoing due to the broad spectrum of possible pharmacological applications of scorpion venom peptide. For example, ABPs have been tested separately or in combination with antibiotics due to their synergistic effect, immunomodulators, and/or endotoxin-neutralizing compounds. In addition, new AMPs from scorpion venoms are currently obtained from traditional purification methods or the designing of synthetic peptides by chemical modifications or mutagenesis of AMPs sequences existing in nature. Based on those above, the peptides’ structure-activity relationship can be determined to enhance the antimicrobial activity of natural APMs and, in the same way, reduce the toxicity against mammalian cells [119,120,121]. Despite the many properties and benefits of AMPs that could be used in the clinic, no results from in vivo studies exist to date. The disadvantages of these peptides are low metabolic stability and oral absorption, rapid excretion through the kidneys and metabolism in the liver, high toxicity and immunogenicity, and production costs. Nonetheless, the FDA approved a few peptide drugs, such as captopril, ziconotide, atracurium, and eptifibatide. Others are undergoing phase clinical or pre-clinical treatment, which indicates that there is still much to be done.

Other studies have focused on the search for peptide drugs with anticancer activity. In the same way as AMPs, venom peptides show broad-spectrum activity on human cancer cell lines. Like resistance to antibiotics, cancer cells are developing high resistance to the existing drugs; thus, developing new anticancer agents with minimized toxicity, side effects, and drug resistance is imperative. The venom of scorpions of the Buthidae family is composed of organic salts, free amino acids, heterocyclic components, peptides, proteins (mainly enzymes), and toxins that affect the function of ion channels, such as sodium (Na^+^), Potassium (K^+^), Calcium (Ca^++^) and Chloride (Cl^−^). These components have the highest concentration [122,123] and are recognized for their cytotoxic capacity on tumor cell lines. Mechanisms attributed mainly to the peptides or toxins found in this secretion; for example, some of them from the venom of *T. serrulatus* have effects on cell lines from the breast, prostate, lung, and glioblastoma [124,125].

In Colombia, there is a great diversity of species of the genus Tityus. *Tityus macrochirus* is the species most abundant in Colombia and causes the most significant number of cases of scorpionism in the country. Therefore, a preliminary study on *T. macrochirus* venom has been carried out to observe the peptide composition and possible activity, managing to determine that it has around 70 peptides, with molecular weights of 3000 to 7800 Da. Venom and isolated peptides have cytotoxic effects on different cell lines from tumors with the highest incidence in the world, such as breast, lung, cervix, colon, and prostate [126]. Besides, these results demonstrated different effects related to the possible activity of ion channels, being the first studies on the structure and function of the peptides present in the venom [127].

## 4. Expectations

With activity research for AMPs, it is possible to increase their activity and stability and improve their pharmacological properties. For example, research is currently being carried out on in vitro e in vivo evaluations of the antimicrobial activity of scorpion venom and its peptides. In addition, many lines of study have been developed to generate new peptide sequences for different molecular targets based on databases of known peptides using computational design. In other cases, studies have focused on optimizing known peptide sequences to make them more resistant to various enzymes or improve their efficiency against cells or proteins [128].

Modern medicine is seriously threatened, and successful surgical interventions today are highly susceptible to complications due to multi-resistant bacteria. Several studies aimed to improve the selectivity and stability of AMPs through chemical modifications, such as shortening or lengthening of the molecules, racemization (enantiomers), and amino acid substitutions in different positions of the molecule [128]. The study of the molecular mechanisms of action of AMPs provides valuable knowledge about the structure/function of, and helps modify or design, antimicrobial peptide sequences for various therapeutic applications [129,130]. It is estimated that microbes will be more resistant to conventional antibiotics shortly [42,131].

## 5. Conclusions

Scorpion venoms have bioactive components against different microorganisms, such as viruses, protozoa, bacteria, and fungi. In addition, some of the compounds are AMPs, which are cationic and exhibit broad-spectrum antimicrobial activity. Libraries can be built from the glands of scorpions using molecular biology tools to get sequences for modifying and enhancing peptide activity. The development of new technological platforms for synthesizing and modifying peptide activity at low cost makes these alternative tools for designing new drugs against multi-resistant strains appealing. Their use is becoming more frequent and successful, even though some of these compounds are still in the early clinical stages. Their application in the medium term represents a promising option, especially if we consider the high drug resistance of many microorganisms.

## Figures and Tables

**Figure 1 toxins-14-00740-f001:**
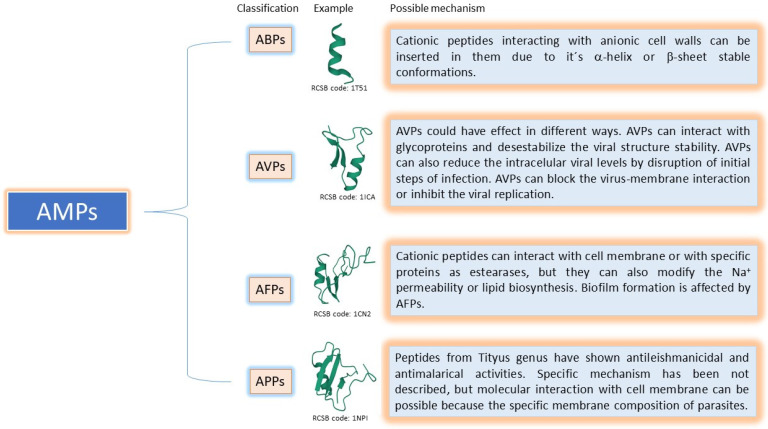
Brief summary of possible mechanisms involved in AMPs’ action.

**Table 1 toxins-14-00740-t001:** Antibacterial activity of peptides isolated from scorpion venoms (ABPs). Some peptides act on a specific microorganism, but other peptides act on several different microorganisms. NR = non reported.

Venom Species	Peptide ABPs	Aminoacids Residues	Molecular Weight (Da)	Antimicrobial Activity	Minimal Inhibitory Concentration (MIC)	Reference
*Opithancatus Madagascariensis*	IsCT1 and IsCT2	13	1504.2 and 1466.0	*S. aureus* and *E. coli*	50 μg/mL	[29]
*Mesobuthus martensii Karsch*	BmKbpp	47	NR	Gram negative bacteria	2.3 to 68.2 µM for different strains	[30]
*Mesobuthus eupeus*	Meucin-49	49	5574.9	Gram negative and Gram positive	0.33 to 8.28 µM and 0.54 to 16.95 µM	[68]
*Heterometrus petersii*	Hp1404	14	1545.9	*Acinetobacter baumannii*	3.2 µM	[31]
*Heterometrus spinifer*	HsAp	29	NR	Gram positive and Gram negative bacteria	11.8 to 46.5 µM and 23.8 to 51.2 µM	[49]
*Pandinus imperator*	Pantinin 1,2 and 3	13–14	NR	Gram positive and Gram negative bacteria	4 to 48 µM and >87 to 36 µM	[48]
*Vaejovis punctatus*	VpAmp1.0 and VpAmp2.0	19 and 25	NR	Gram positive and Gram negative bacteria	2.5 to 15 µM and 2.5 to 24 µM	[34]
*Scorpiops tibetanus*	StCT2	14	NR	*Staphylococcus aureus*	6.25 µg/mL	[27]
*Tityus serrulatus*	TsAP-2	17	NR	*Staphylococcus aureus*	5 µM	[46]
*Mesobuthus martensii*	Marcin-18	18	2134.3	Gram positive and Gram-negative bacteria	1.5 to 23.4 µM and 5.9 to 11.27 µM	[69]
*Tityus stigmurus*	Stigmurin	17	NR	*S. aureus*	1 µg/mL	[70]

**Table 2 toxins-14-00740-t002:** Peptide from different scorpion venoms have shown antifungal activity (AFPs). The table shows the genus, an antifungal activity, in different types of fungi, with a specific minimum inhibitory concentration (MIC). NR = non reported.

Species Venom	Antifungal AFPs	Aminoacids Residues	Molecular Weight (Da)	Inhibited Fungal Species	Minimal Inhibitory Concentration (MIC)	Reference
*Mesomexovis variegatu*	VpCT1	13	1463.7	*C. albicans* and *C. glabrata*	25 µM and 12.5 µM	[34,52,82]
*Mesomexovis variegatu*	VpCT2	13	1535.7	*C. albicans* and *C. glabrata*	25 µM	[34,52,82]
*Androctonus australis*	Androctonin	25	3076.7	*Aspergillus brassicola*, *Stemphylium*, *Fusiarum culmorum*, *Botritis cinérea*	2 a 50 µM	[83]
*Tityus stigmurus*	Stigmurin	22	NR	*C. albicans*, *C. krusei* and *C. glabatra*	34.75; 69.5 and 69.5 µM	[84]
*Tityus stigmurus*	Hypotensin (TistH)	25	2700	*C. albicans*, *C. tropicalis*, *A. flavus* and *Trichophyton rubrum.*		[85]
*Tityus stigmurus*	Hypotensin (TistH)	25	2700	*C. albicans*, *C. tropicalis*, *C. parapsilosis*, *C. glabrata*, *C. krusei*, *C. dubliniensis*, *C. rugosa*	>178.5 µg. mL^−1^	[86]
*Parabuthus schlechteri*	Parabutoporin	45	NR	*Neurospora crassa*, *B. cinerea*, *F. culmorum*, *S. cervisiae*	2.5; 3.5; 0.3; 2 µM	[67]
*Opistophtalmus carinatus*	Opistoporin 1	34	NR	*N. crassa*, *B. cinerea*, *F. culmorum*, *S. cervisiae*	0,8; 3.1; 0,8; 2 µM	[67]
*Tityus obscurus*	ToAP2	26	NR	*C. albicans*, *C. tropicalis*, *C. parapsilosis*, *C. glabrata*, *C. neoformans.*	12.5; 3.12; 50; 200; 12.5 µM	[87]
*Tityus obscurus*	ToAP3	17	NR	*C. albicans*, *C. tropicalis*, *C. parapsilosis*, *C. glabrata*, *C. neoformans.*	25; 12.5; 200; 200; 50 µM	[87]
*Tityus obscurus*	ToAP1	17	NR	*C. albicans*, *C. tropicalis*, *C. parapsilosis*, *C. glabrata*, *C. neoformans.*	50; 12.5; 200; >400; 25 µM	[87]
*Opisthacanthus cayaporum*	Con10	27	NR	*C. albicans*, *C. tropicalis*, *C. parapsilosis*, *C. glabrata*, *C. neoformans.*	100; 12.5; 200; 200; 50 µM	[87]
*O. cayaporum*	NDBP-5.7	17	NR	*C. albicans*, *C. tropicalis*, *C. parapsilosis*, *C. glabrata*, *C. neoformans.*	25; 25; >400; >400; 25 µM	[87]

**Table 3 toxins-14-00740-t003:** Peptides from different scorpion venoms that have shown antiviral activity (AVPs). The table shows the genus of scorpion, together with the peptide that performs antiviral activity, in different types of viruses. NR = non reported.

Species	AVPs	Amino Acid Residues	Molecular Weight (Da)	Inhibited Virus	Concentration Required to Reduce Virus Infection by 50% (EC_50_)/Activity	Reference
*Heterometrus petersii*,	Hp1090	13	NR	HCV	7.62 µg/mL, permeabilize the viral envelope and thus inactivate.	[91,105]
*H. petersii*	Hp1036 and Hp1239	13	NR	HSV-1	0.090 and 0.06 µM, respectively. Inhibit several steps of the replication cycle	[91,106]
*Chaerilus tryznai*	Ctry2459	9	NR	HCV	1.84 µg/mL, inhibits the initial infection	[91,107,108]
*Lychas mucronatus*	Mucroporin M-1	17		SARS-CoV and H5N1	7.12 and 1.03 µM, respectively	[101,109]
*Euscorpios validus*	rEv37	18	8503.96	HSV-1, DENV-2, HCV and ZIKV	Reduces infection at the post-entry cycles of infection.	[101,110]
*Euscorpios Validus*	Eval418	13	NR	HSV-1	2.48 µg/mL	[99,111]

**Table 4 toxins-14-00740-t004:** Peptides from scorpion venoms have shown antiparasitic activity (APPs). The table shows the genus of the scorpion, together with the peptide that performs an antiparasitic activity, in different types of parasites. NR= non reported.

Species	APPs	Amino Acids Residues	Molecular Weight (Da)	Types of Parasites	The concentration Required to Reduce Grown of Parasite	Reference
*T. stigmurus*	StigA25 and StigA31	Both 17	NR	Epimastigote forms of T. *cruzi*	12.5 μM and 25 μM respectively	[116]
*T. serrulatus*	Pep 1 and Pep 2a	11 and 10	1467 and 1509	*Toxoplasma gondii*	50 μg/mL	[117]
*Hoffmannihadrurus gertschi*	rHge36	48	NR	*T. crassiceps*	67 nM	[118]
*P. imperator*	scorpine	75	8350	*Plasmodium*	Inhibited the fecundation and ookinete, with concentration of 50 and 3 µM respectively.	[38]

## Data Availability

Not applicable.

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
