# Peer review of "Antimicrobial Activity Developed by Scorpion Venoms and Its Peptide Component"

_toxins, 2022, doi:10.3390/toxins14110740_

Round 1

Reviewer 1 Report

The review entitled “ANTIMICROBIAL ACTIVITY DEVELOPED BY SCORPION  VENOMS AND ITS PEPTIDE COMPONENT” summarized many study describing antimicrobial potential of scorpion’s venoms and their peptides. I have few comments that should be taken into consideration in this context.

1.      Please revise the abstract and underline exactly the tasks of the present research. It is not so clearly reported why the present research it is important for the readers and present needs. please also correlate with the introduction and conclusion sections.

2.      The name of bacterial strains in the manuscript should be italic representation. Starting from introduction sections.

3.      Reading the main goal of the paper described in the introduction section I suggest to revise the title and tasks ones again. Please correlate this aspect.

4.      Section 2.1. a) the author describe the possible mechanism of AMPs taking place. Therefore, to attract more attentions to the respective paper I suggest to design a known mechanisms reported already. Or design your own summarizing all reported mechanisms. Additionally, author should chose or design the mechanism of each antimicrobial sections separately or design for section 2 one mechanism scheme including antibacterial, antifungal, antiviral and antiparasitic activity.  

b) the title of the section is antimicrobial, however, the table 1 present only results regarding the antibacterial properties. Moreover, next section is dedicated to the bacterial strains.

5.      Section 2.2. the authors dedicated this section to bacteria species however, the section is started with the antimicrobial potential. The author should clarify and differentiate the antimicrobial and antibacterial properties if want to underline one aspect. Create misunderstanding reading section 2.1 and 2.2; antimicrobial is too general while antibacterial is addressed strictly to  the bacteria. Please revise this aspect.

6.      Moreover, the author designed the manuscript and divided the sections showing the separate anti- bacterial, fungal, viral and parasitic effect. However, the sections should be restructured.

Proposed example:

 section 2: Scorpion venom – antimicrobial case report.

Section 2.1: antibacterial properties

Section 2.2. antifungal etc…..

Moreover, the author not clearly described the full name of AMPs abbreviation. If the author mean such abbreviation as a Antimicrobial peptides (AMPs) should be clearly specified.

Additionally, next sections authors described antibacterial effect of AMPs. How this to understand – antibacterial effect of antimicrobial peptides? This point should be revised . if you want to show antimicrobial should be (AMPs), if you mean antibacterial should be (ABPs) etc….Similarly,  in case of all antiviral and antiparasitic activity. Such representation create misunderstanding.

7.      Please also revise the structure of the tables. Or present one table representing all antimicrobial properties or create for each section separate table. Additionally, please disscuse more literature for antiviral and antiparasitic effect section.

8.      The author described in the introduction section the problem of microorganisms; I did not find any arguments regarding the antifungal, antiviral etc…. as is presented in the manuscript. Please clarify this. The author should structure based on the microbe classification.

Reviewer 2 Report

The topic is very interesting, but the article lacks depth in its approach. The authors present an extensive list of antimicrobial agents derived from scorpion venoms, however, it does not present a clear perspective of the therapeutic potential of these agents as is their objective. The work needs to mature to better organize the information presented and offer a greater contribution to the understanding of the possibilities of using these substances, their advantages and limitations. Good spelling and grammar checking are required. Some specific observations follow:

In section "1. Introduction", species names are not represented graphically in italics. A review is needed.

In line 29, the abbreviation “AMPs” was not previously mentioned.

There is a problem with the representation of Latin characters in reference 5 (lines 453-455).

Regarding the information provided in the introduction: “Due to 29 antibiotic-resistant infections, annual deaths worldwide are expected to increase. from 700,000 in 30 2014 to 10 million in 2050 [4, 5]", the original data source was not cited in the reference, reference 4 is citing another citation.

There is a grammatical problem with the sentence “…found in the venom of scorpions belonging to both the Buthidae family and the other families.”

In section 2. Scorpion venom, the paragraph is too long and a little confusing. I suggest re-rewrite the text.

The authors cite some toxins and report that they stand out, but do not explain why (lines 65-68).

In line 70 there seems to be a misinterpretation of the cited reference, the authors in the cited reference talk about the role of AMPs in invertebrates and not in unicellular or mammals and therefore it is not appropriate to substantiate the affirmation.

Line 111: changes “The first scorpion was isolated…” to “The first scorpion toxin was isolated…”

The text in lines 162-164 is confusing, the authors sometimes speak in three and sometimes in four compounds. What is correct?

Lines 160-189, perhaps the data present in this paragraph could be presented in a table. The same applies to lines 190-227. As the text is just a list of various toxins and their antimicrobial activity, I suggested that a table be presented instead and in the text, only relevant remarks should be made.

There is an abbreviation error on lines 236, 237 and 272.

The first paragraph of the discussion is a repeat of the first one of the Introduction.

In the introduction, the authors say that AMPs are amphipathic, in the discussion it is said that it is hydrophobic. This problem needs to be fixed.

In Section 4. Expectations, what are PRONAUDCA and GRIP? What is your relationship with the review article?

On line 406 there is a word written in Spanish.

Reviewer 3 Report

The manuscript, which title is “ANTIMICROBIAL ACTIVITY DEVELOPED BY SCORPION VENOMS AND ITS PEPTIDE COMPONENT”, is interesting. However, there are several questions in the manuscript. First, the world format of bacteria and animal source should show italics in main text. There are several typo and grammar mistakes. The authors should provide the information about the toxic concentration in the manuscript. In addition, the authors should provide the detail mechanism of the Peptide from different scorpion venoms in the new table.

Reviewer 4 Report

All comments are inserted in pdf format.

The abstract, title and conclusions are not covering all subjects ( e.g. venom antiviral, antifungal and antiparasitic properties).

The references should be distributed inside the text. It is necesarly to avoid in block citations where is possible.

Reviewer 5 Report

The review paper entitled “Antimicrobial Activity Developed by Scorpion Venoms and Its Peptide Component” describes multiple studies carried out with the use of scorpion venoms against pathogenic bacteria, highlighting the high therapeutic potential of the different bioactive molecules produced by them. Although of interest for the scientist in this specific filed this is a bit basic overview of the topic that is a bit hard to follow due to extremely long sentences.

There is no search strategy explained in the paper. How did authors select papers to be included in this review? Which data bases were searched and with what keywords. What were including/excluding criteria/parameters? This is very important for a good literature overview. Besides, what is the type of this review paper… narrative systematic, scoping etc.?

Authors could take a look on a newly published paper by von Reumont and co-workers especially in regard to pharmaceutical application of venom peptides in antimicrobial research.

von Reumont et al. (2022) Modern venomics-Current insights, novel methods, and future perspectives in biological and applied animal venom research. 18;11: Giga Science. doi: 10.1093/gigascience/giac048.

I would suggest to authors to discuss possible toxicity towards normal non-target cells since such venoms and their constituents are cyto/genotoxic to normal cells as well. Although there are numerous animal venoms and their components that often show good results towards cancerous cells, bacteria, viruses as well as other disorders and diseases, there are always open questions regarding their potential toxicity on normal non-target or host cells and tissues making this kind of toxicity one of the biggest obstacles for the possibility of applying such natural products as medications.

Minor remarks:

Omit one “thus” from the first sentence in the Abstract section.

There are several too long sentences that could be split in shorter ones to have a better flow (e.g. first sentence in the Abstract that has 5 lines, first and second sentences of the second paragraph of the Introduction section etc. – this all could be split and shortened for an easier reading)

Put Latin names in italic font. Additionally, on first mention whole Latin names of specie should be provided.

Page 1, line 22 – put full stop after the worldwide and start new sentence.

Page 1, line 29 – full word for AMPs… pay attention on abbreviations throughout the paper.

Usually after the semicolon there is no need for a capital letter.

Table 1 – “from scorpion venoms” (there are many spelling mistakes that should be corrected)

Page 7, line 252 – “three replicates). by means” – please correct

Put in vitro in italic

Round 2

Reviewer 1 Report

Thank you the author for the revision, now the paper is more closed to the reality. However,  I would ask the authors two improve additionally few aspects:

1. I would like to encourage the author once again  to revise the introduction section and discuss deeply the topic of antimicrobial peptides (AMPs). Please take in considerations each peptides properties (ABP, AVP, AFP and APP) and debate the novelty.  Poorly is presented in this form.

2. please also correlate the abstract and conclusion and main goal of the study and present clearly the problem. In such way is poorly discussed and more general. E.g author used in the abstract section “Different bioactive molecules”  - I think should be detailed this point. Which exactly the bioactive molecules have been discussed and why?  

3. I suggest once again to discuss more the sections AVP and APP with newest reports.  

Reviewer 3 Report

none

Author Response

Thank you for your suggestions to the amnuscript and your acceptance.

Reviewer 4 Report

 The paper can be accepted without any further changes.

Author Response

(The authors gave the same response as above.)

Reviewer 5 Report

Although authors did not respond explicitly to any of my raised questions, the manuscript is improved and could be accepted for publication.

Author Response

(The authors gave the same response as above.)
